# How to Treat Melanoma? The Current Status of Innovative Nanotechnological Strategies and the Role of Minimally Invasive Approaches like PTT and PDT

**DOI:** 10.3390/pharmaceutics14091817

**Published:** 2022-08-29

**Authors:** Joana Lopes, Cecília M. P. Rodrigues, Maria Manuela Gaspar, Catarina Pinto Reis

**Affiliations:** 1Research Institute for Medicines, iMed.ULisboa, Faculty of Pharmacy, Universidade de Lisboa, Av. Professor Gama Pinto, 1649-003 Lisboa, Portugal; 2Instituto de Biofísica e Engenharia Biomédica, Faculdade de Ciências, Universidade de Lisboa, Campo Grande, 1749-016 Lisboa, Portugal

**Keywords:** melanoma, treatment, nanotechnology, drug delivery, photothermal therapy, photodynamic therapy, regulation, clinical trials

## Abstract

Melanoma is the most aggressive type of skin cancer, the incidence and mortality of which are increasing worldwide. Its extensive degree of heterogeneity has limited its response to existing therapies. For many years the therapeutic strategies were limited to surgery, radiotherapy, and chemotherapy. Fortunately, advances in knowledge have allowed the development of new therapeutic strategies. Despite the undoubted progress, alternative therapies are still under research. In this context, nanotechnology is also positioned as a strong and promising tool to develop nanosystems that act as drug carriers and/or light absorbents to potentially improve photothermal and photodynamic therapies outcomes. This review describes the latest advances in nanotechnology field in the treatment of melanoma from 2011 to 2022. The challenges in the translation of nanotechnology-based therapies to clinical applications are also discussed. To sum up, great progress has been made in the field of nanotechnology-based therapies, and our understanding in this field has greatly improved. Although few therapies based on nanoparticulate systems have advanced to clinical trials, it is expected that a large number will come into clinical use in the near future. With its high sensitivity, specificity, and multiplexed measurement capacity, it provides great opportunities to improve melanoma treatment, which will ultimately lead to enhanced patient survival rates.

## 1. Introduction

Today, cancer is undoubtably one of the biggest public health problems around the world. According to the World Health Organization, it is the leading cause of death in people under 70 years old in 57 countries, and the second-leading cause in another 55 countries [1]. Along with cancer, cardiovascular diseases also have a great impact on global mortality [2]. However, although today these diseases are still the leading cause of premature death in 70 countries, the predicted decrease in their mortality rate indicates that cancer will surpass them as the leading cause of death in many countries [1,2]. The International Agency for Research on Cancer (IARC) estimated that in 2020, cancer affected around 19.3 million new people and claimed the lives of around 10 million worldwide. Furthermore, the projections are not encouraging. The growth trend of recent years with regard to the number of new cases is expected to continue and an increase of almost 50% is predicted for 2040, totaling 28.4 million cases [1].

In this context, skin cancers are no exception, and among them, melanoma, despite having a much lower incidence compared to others, is the most aggressive form, responsible for most skin cancer-related deaths [3,4,5]. The IARC estimates that in 2020 there were about 325,000 new cases and 57,000 deaths associated with cutaneous melanoma, and increases of more than 50% for both indicators are expected within 20 years [6,7]. Air pollution, depletion of the atmospheric ozone layer and global warming are some of the factors that contribute to this increase [8,9].

Melanoma is characterized as a complex and unpredictable malignancy that derives from the malignant transformation and consequently uncontrolled proliferation of melanocytes [3,10]. Melanocytes are melanin-producing cells, which, in addition to providing color to the skin, play an important role in protecting it from ultraviolet radiation [11]. Its location is predominantly in the basal layer of the epidermis [8,9], and therefore, cutaneous melanoma comprises the vast majority of diagnosed cases [5,12]. However, melanocytes can still be present in a wide variety of other tissues, such as hair follicles, the eye, the inner ear, the mucosa surfaces, or even the central nervous system. These cases comprise the non-cutaneous forms of melanoma [9,10,11,12].

In terms of therapeutic strategies currently available for the treatment of melanoma, surgery, radiotherapy, chemotherapy and, more recently, immunological and targeted therapies are included [13]. The approval of these last two therapeutic classes has completely revolutionized the therapeutic arsenal. However, these therapies are still far from being considered satisfactory [14]. The resistance and significant adverse side effects, together with the complexity, and consequent high morbidity and mortality associated with melanoma, continue to capture the interest of the scientific community. The investment in the search for greater knowledge about the underlying pathophysiological mechanisms, as well as the development of more selective, effective and safer treatments, still remains a priority [15,16]. Herein, nanotechnology appears as a potential strategy to improve treatment outcomes while reducing adverse side effects. The use of lipid-based systems, polymeric, metallic, and hybrid nanoparticles (NPs) as carriers of one or more anticancer agents as well as its possible combination with physical methods, such as photothermal (PTT) and photodynamic therapies (PDT), have shown promising results in this field [17,18,19,20]. Thus, this review aims to provide an updated approach to the latest nanotechnological advances as innovative therapeutic strategies in the management of melanoma by using different types of nanosystems, as schematically represented in Figure 1.

## 2. Melanoma

As already reported, melanoma comprises a malignant transformation of melanin-producing cells. This transformation is a highly complex and multifactorial process that arises from the interaction of several modifiable and non-modifiable risk factors [12]. Exposure to ultraviolet radiation, either through sunlight or the use of tanning devices, is the modifiable risk factor with the greatest impact on the number of new diagnosed cases of cutaneous melanoma, being responsible for 60–70% of cases [21,22]. In addition, sunburn at an early age, certain medications, such as immunosuppressive drugs, or environmental exposure to some chemicals, such as pesticides or heavy metals, complete the range of modifiable risk factors [3,23]. Regarding non-modifiable risk factors, age, sex, ethnicity, a high number of common nevi or atypical nevi, a personal or family history of skin cancer, diseases that compromise the immune system, genetic alterations, specific genetic conditions such as albinism and even some individual phenotypic traits such as the presence of freckles or light eyes and skin are all factors associated with an increased risk of melanoma [12,23,24,25]. Skin color and its response to UV radiation are parameters that allow the categorization of different skin phototypes. The Fitzpatrick scale is the most used numerical classification for human skin color. Herein, skin types vary from type I to VI. The higher the phototype, the lower the associated risk of melanoma [5,24].

In addition to the etiological heterogeneity mentioned above, the clinical presentation is also quite variable, with distinct epidemiological, dermatological, and histopathological characteristics associated with each subtype. Among the main types of cutaneous melanoma, and in increasing order of incidence, are superficial spreading melanoma, nodular melanoma, *lentigo maligna* melanoma, and acral lentiginous melanoma [4]. Therefore, the selection of the most suitable therapeutic option takes into account, among other individual aspects, the anatomical location and the stage and genetic profile of the tumor [13]. The five-year overall survival of a patient with advanced stage melanoma does not exceed 15–20%. On the contrary, if detected at early stages, this value increases exponentially to 99% [24]. These numbers emphasize the importance of a combined strategy in the prevention and early detection of this type of malignancy to improve the prognosis and survival rates of patients [26,27].

## 3. Nanotechnology Applied to Melanoma Therapy

Nanotechnology is a concept created in 1959 by the Nobel Prize laureate in Physics, Richard Feynman. Since then it had an extraordinary evolution, being even considered one of the most promising technologies of this century [28]. Nanotechnology comprises the design, development, production, characterization, and application of materials at the nanoscale [29,30]. The definition of nanoparticle is not completely consensual, because although some authors defend its variation between few nanometers and 100 nm, sizes up to 1000 nm can also be considered in drug-delivery fields [31,32,33,34,35].

The application of nanotechnology to medicine, called nanomedicine, has aroused growing interest from researchers around the world. It is present not only in the treatment of various diseases, but also in diagnosis and monitoring, as well as in immunization and vaccine development [20,36,37,38,39].

The behavior NPs in in vitro and in vivo systems, and consequently the rate of drug release, is closely related to several physicochemical characteristics of the nanosystems. Adhesion to the cell surface, phagocytosis and degradation profile are dependent on the particle size, charge, morphology and surface chemistry of the NPs as well as on the hemodynamic properties and density of the drug binding sites [40,41].

Concerning the drug release from nanodelivery systems, it generally occurs through different mechanisms, depending on how the drug is associated within the nanosystem, through covalent linkage at their surface, incorporated in the matrix, following complexation, via interaction or physical encapsulation of unmodified drug molecules. Regarding the cleavage of covalently conjugated drugs, this can be done by ester, amide, or hydrazone hydrolysis, disulfide exchange, hypoxia activation, self-immolation reaction (enzyme, thiol-disulfide exchange, low pH and light-promoted), photochemistry and thermolysis [41,42]. On the other hand, non-covalently bound drugs are released by controlling the pore size, the effective volume of the carrier as well as the internal conformation and permeability. Furthermore, the release of associated drugs can be promoted by external stimuli such as pH, temperature, and use of ultrasound [41,42]. Moreover, the type of coating and the thickness of the encapsulating material affect the release rate [41].

Additionally, the fate of NPs in the human body might also raise some safety issues. In general, NPs intravenously administered circulate in the bloodstream until they are cleared and eliminated. Renal and hepatobiliary elimination are the two main mechanisms. The pathway is different between biodegradable nanocarriers and nonbiodegradable NPs like gold nanoparticles (Au NPs). It has been shown that small metallic NPs (smaller than 5.5 nm) can be mostly cleared by renal filtration and urinary excretion and, in some studies, by feces. Larger NPs can prevent the renal excretion and can be accumulated in the liver and spleen. Several studies have suggested that nanomaterials can be eliminated in a hepatobiliary manner, through transcytosis in the hepatocytes, resulting in transport to the biliary system, followed by the gastrointestinal tract, and eventually eliminated in feces. In this case, hepatobiliary elimination is normally slow, ranging from hours to months or even longer. The liver sinusoidal endothelial cell fenestrae size in humans was reported to be 107.5 ± 1.5 nm [43]. Particles with dimensions that are larger than these fenestrae cannot directly enter the space of Disse, but may access it by a less efficient and slower process via transcytosis through the liver sinusoidal endothelial cells and Kupper cells [44].

In terms of treatment, nanotechnology aims to overcome some of the drawbacks of current therapies referred to in the Introduction section. The protection of sensitive therapeutic agents, improvement of pharmacokinetic and pharmacodynamic profiles and, consequently, bioavailability, as well as an increase in selectivity for tumor cells either by passive or active targeting, are some of the promises of this type of nanosystem. Thus, an increase in the safety and efficacy of treatments is expected [45,46,47,48,49]. The success and added value of nanomedicine is confirmed by the various nanomedicines that have already been approved or are undergoing clinical trials [50,51]. The first nanomedicine approved by the Food and Drug Administration (FDA) and European Medicines Agency (EMA), in 1995 and 1996, respectively, was Doxil^®^, precisely in the area of oncology. Doxil^®^ is a liposomal delivery system of doxorubicin commonly used in the treatment of Kaposi’s sarcoma, multiple myeloma, metastatic breast cancer, and ovarian cancer [52]. Since then, several types of nanosystems have been explored for potential application in melanoma. They include systems acting either as carriers of natural compounds, new or existing drugs and genes, or as antimelanoma agents per se. Moreover, the nanosystems can even be activated by external stimuli, such as PTT and PDT [45,53,54,55,56,57,58].

Although phototherapies have experienced an increased interest in recent years [59], the use of light as a therapeutic strategy has already started thousands of years ago [60]. The therapeutic use of sunlight, denominated heliotherapy, is the oldest phototherapeutic strategy and, until the mid-19th century, the only [60] type. Furthermore, for more than 3000 years, ancient Egyptian, Chinese, and Indian civilizations have combined heliotherapy with reactive chemicals to treat a wide range of skin conditions, such as psoriasis, vitiligo or cancer [61,62,63]. At the end of the 19th century and beginning of the 20th century, discoveries made by different researchers such as Hermann von Tappeiner, Oscar Raab, or Niels Finsen paved the way for modern phototherapy [61,64]. In 1960, with the invention of the laser by Theodore H. Maiman, the ablation of tumors through the use of lasers began to be explored. However, the lack of selectivity and the high power densities required raised safety concerns [64]. Phototherapies such as PTT or PDT associated with laser irradiation and involving the administration of photosensitizing agents, are able to trigger a series of chemical, biological, and physiological reactions that culminate in the death of neoplastic cells [65].

PTT is based on the increment of temperature at the tumor site (which can range from 41 to 49 °C) through photothermal enhancers capable of converting the optical energy received from the laser into thermal energy, thus taking advantage of the low resistance of cancer cells to heat. Depending on the temperature reached, mechanisms of apoptosis, necrosis and necroptosis are pointed out [59,66]. In its turn, PDT acts by the chemical damage promoted in the cells, through the production of reactive oxygen species (ROS) due to the activation of a photosensitizing agent by irradiation. An activation of signaling pathways such as apoptosis and necrosis are described. In addition, there is also a collapse of the blood vessels surrounding the tumor, compromising irrigation, also inducing cell death by hypoxia [61,64,65,67]. Moreover, both PTT and PDT have demonstrated a slight ability to stimulate an immune response [68,69].

Although photothermal agents have not yet been tested in large clinical trials and only laser ablation is currently used in clinic, PDT has been applied in the cancer field for over 40 years [64]. The advantage of phototherapies is based on its high efficiency and selectivity in the ablation of cancer cells and minimal invasiveness that allows the reduction of adverse effects and a faster recovery of the patient [65,70,71,72]. However, limited laser penetration restricts phototherapies to the treatment of superficial cancers [72,73]. The additive and synergistic effect of combining PTT or PDT with other therapeutic approaches is increasingly being investigated [19,38,64,70,73,74,75,76,77].

In the following sections, different types of nanotechnology-based strategies are discussed. In melanoma management, several types of nanosystems, as depicted in Figure 2, have been used. Lipid-based systems, such as liposomes, solid lipid nanoparticles (SLNPS) and nanostructured lipid carriers (NLCs), and polymeric, metallic, and hybrid nanoparticles are some examples listed in the present review. The reported outcomes are based on data published by the authors in international journals and do not reflect a personal opinion.

### 3.1. Lipid-Based Systems

Lipid-based drug-delivery systems are gaining more and more notoriety, either by improving currently available therapies, or by creating new ones. This happens as a result of their ability to transport and allow a controlled release of the most varied molecules. Their high stability, biocompatibility, and biodegradability have been also contributive factors for their widespread use [78,79]. Among these usages, liposomes as vesicular systems or lipid particulate systems such as SLNPs and NLCs have attracted special interest [80]. In the following section, each system and respective in vitro and in vivo examples will be briefly described. Several human and murine melanoma cell lines have been used. In particular, one of the most studied is the murine melanoma B16F10 cell line. This cell line exhibits a morphology of spindle-shaped and epithelial-like cells that was isolated from skin tissue of a mouse with melanoma [15,81]. These cells are highly metastatic and present strong pigmentation. Furthermore, in vivo studies are most often tested in C57BL/6 male mice; that is, the mice strain in which the melanoma cell line B16F10 was created. This strain is widely used in immunocompetent murine melanoma models [15,81,82]. These and more in vitro and in vivo examples are described in Table 1, Table 2 and Table 3.

#### 3.1.1. Liposomes

Liposomes, as drug delivery, are the most well-known, versatile, and represented lipid-based system clinically approved for the treatment of several diseases [83]. The first application in the field of cancer was initiated by Gregory Gregoriadis in the early 1970s [84,85].

Liposomes are spherical vesicles constituted essentially by phospholipids, which are organized in lipid bilayers separated by aqueous compartments. This structure guarantees the loading of hydrophilic drugs in the aqueous compartments and hydrophobic molecules in the lipid bilayers. In addition to phospholipids, liposomes can also include other constituents, such as glycolipids, cholesterol and hydrophilic polymers such as polyethylene glycol (PEG) [86,87]. The latter is widely used to prolong the circulation time of the formulation in the bloodstream. PEG prevents liposome uptake by the reticuloendothelial system, potentiating its extravasation and accumulation at tumor sites through the permeability and retention (EPR) effect [88].

There are some studies reporting the use of liposomes for melanoma management [89,90,91,92,93,94,95,96,97,98,99,100,101,102,103]. A broadly used strategy to improve efficacy has relied on the simultaneously delivery of different liposomal formulations, pursuing a synergistic effect. In this context, Saqr and collaborators [89] proposed the in vitro cytotoxicity assessment of combining hispolon and doxorubicin (DOX) liposomes. DOX is a well-known chemotherapeutic agent with marked adverse side effects. In turn, hispolon, a polyphenol from a natural source, has noticeable in vitro cytotoxic activity, namely in melanoma [104]. Their combination aimed to potentiate the anticancer activity of DOX, while reducing effective dosage and consequently adverse side effects. In vitro assays on the murine melanoma cell line B16BL6 showed a significantly higher cytotoxicity of the conjugation of developed formulations when compared to hispolon-loaded liposomes or DOX-loaded liposomes individually. Furthermore, a significantly higher percentage of in vitro apoptotic cells was observed for the combined strategy, in comparison to control, hispolon-loaded liposomes, and DOX-loaded liposomes.

The application of metal-based complexes in the treatment of cancer has also become relevant [105]. One example is the work developed by Nave et al. and Pinho et al. [92,98], which designed and physicochemically characterized different, long-circulating liposomes incorporating a copper complex for melanoma management, the 1,10-phenanthroline ligand (Cuphen). First, its antiproliferative properties toward tumors cells as well as in vivo safety profile were demonstrated [92]. Liposomal formulations with suitable Cuphen loadings were selected for in vitro and in vivo studies. In vitro antiproliferative properties of copper complex were preserved after incorporation in liposomes. In a murine melanoma model, two selected liposomal formulations significantly reduced tumor progression, and no toxic side effects were observed compared to the control group or animals treated with Cuphen in the free form [98].

In addition to passive targeting via EPR effect, as mentioned above, the active targeting is another strategy whose objective is to enhance the selectivity of the formulation toward tumor sites [106]. In this segment, the group of Merino took precisely this approach and decorated the surface of empty liposomes with monovalent variable fragments (Fab’) of α-PD-L1 [93]. This ligand, in addition to promoting the targeting of the formulation, will also have an immunotherapeutic effect. Thus, after optimizing the immunoliposome production method, DOX was added as a chemotherapeutic agent to the formulation, and its in vitro and in vivo evaluation was performed [99]. Empty liposomes (lip), non-targeted DOX liposomes (DOX lip) and immunoliposomes (Fab’-anti-PD-L1 DOX lip) were tested with respect to cellular uptake efficiency and cytotoxicity in the murine cell line B16OVA, which is known to overexpress PD-L1. Results demonstrated that, in short periods of time (4 h), immunoliposomes significantly increased cellular uptake, and, consequently, its cytotoxicity. An approximately 30-fold decrease in the IC_50_ value was observed compared to non-targeted DOX liposomes. In the same work, intravenous administration of a single dose or three cycles every 72 h were performed to evaluate the extent of the immune response and the antitumor activity, respectively. C57/BL6 mice were inoculated with the same cell line tested in vitro, and then they were divided into five different groups based on the treatment: control group, DOX in its free form, DOX lip, co-administration of DOX lip with 28 µg/mice of free α-PD-L1, and the developed immunoliposomes. The concentration of DOX, 3 mg/kg, was maintained constant in all formulations. Immunoliposomes have been shown to induce a local and systemic immune response compared to the control group. In addition, a more pronounced and sustained antitumor effect over time, as well as a longer survival rate, were also achieved compared to all other treatment groups. Furthermore, immunoliposomes extended the animals’ life expectancy by almost 1 month versus untargeted formulation.

As reported throughout the work, the combination of two or more different therapeutic strategies has been widely investigated in order to enhance the treatment outcomes [107]. In this sense, Xu and collaborators also investigated the potential of a liposomal formulation that combines immune and photothermal therapies [91]. The developed approach consists of thermally responsive liposomes containing the photoactive dye indocyanine green (ICG) and the immune stimulatory molecule polyinosinic:polycytidylic acid (poly I:C). First, the photothermal efficiency as well as the successful release of poly I:C from the formulation was proven. Next, after a preliminary in vitro efficacy assessment on CT-26 murine carcinoma cells, an in vivo model of melanoma was conducted. This model was divided into two phases. First, subcutaneous inoculation of B16F10 cells into C57BL/6 mice was performed to simulate a primary tumor model. At this point, with appropriate size tumors, empty liposomes, ICG-loaded liposomes, and co-loaded ICG and poly I:C liposomes were intratumorally administrated, and the application or absence of 5 min of PTT was tested. Later, the same animals were subjected to a metastatic model by inoculating the same cell line but intravenously. In terms of results, the combination of PTT with the administration of ICG liposomes or the liposomes co-loaded with ICG and poly I:C promoted anticancer effects. However, poly I:C released from the combination therapy was essential for the activation of tumor antigen-specific immune responses by dendritic cells, enabling the prevention of lung metastases and, consequently, a high survival rate of the animals. These examples and some more are briefly described in Table 1.

**Table 1 pharmaceutics-14-01817-t001:** In vitro and in vivo assessment of the therapeutic potential of different liposomal formulations in melanoma models.

Nanosystem Composition	Compound(s)	Model(s)	Summary of Experimental Assays and Conditions	Main Conclusions	Reference
DSPC:chol:DSPE-PEG	Hispolon or DOX	In vitro: murine B16BL6 cell line	Cell viability assay: MTT (0.5 to 50 μM of hispolon and up to 1 μM of DOX) Cell uptake assay: coumarin-6 Cell death assay: Annexin V/PI	Combination of hispolon and DOX liposomes exhibited a greater cytotoxicity compared to their use alone.	[89]
SPC:chol:DSPE-PEG	Ruthenium (II) triazolopyrimidine complex	In vitro: human A375 and Hs294T cell lines	Cell viability assay: MTT (0.4 to 2.5 μM)	Liposomes allowed a reduction of more than 10-fold in the IC_50_ value in relation to its free form.	[90]
DOTAP:chol:C8-ceramide:DSPE-PEG	DOX	In vitro: murine B16BL6 cell line	Cell viability assay: MTT (0.5 μg/mL of DOX and 10 mg/mL of C8-ceramide)	Co-delivery of DOX and C8-ceramide in a liposomal formulation displayed higher cytotoxicity in comparison with DOX liposomes without ceramide or even DOX solution.	[96]
HSPC:DSPE-PEG	Curcumin	In vitro: human MUG-Mel2 cell line	Cell viability assay: MTT (5 and 10 μM) without and with PDT application (380–500 nm; 2.5 J/cm^2^, 2 min) Cell proliferation assay: wound healing Cell death assays: annexin V-FITC/7-AAD and immunocytochemistry	After PDT treatment, curcumin liposomes demonstrated increased phototoxicity and decreased motility in melanoma cells. In turn, in healthy cells (HaCat), a reduced toxicity was observed.	[97]
DMPC:chol:DSPE-PEG or DMPC:CHEMS:DSPE-PEG	Copper (II) complex—Cuphen	In vitro: murine B16F10 cell line	Cell viability assay: MTS (0.2 to 12 μM)	In vivo assays demonstrated the safety and high impairment of tumor progression of Cuphen liposomal formulations in comparison to the free form.	[98]
		In vivo: male C57BL/6 mice; s.c. injection of B16F10 cells	i.v. injection of Cuphen in free form and loaded in two different liposomal formulations (2.5 mg/kg)		
HSPC:chol:DSPE-PEG	DOX and Fab’-anti-PD-L1	In vitro: murine B16OVA cell line	Cell viability assay: SRB (0.001 to 100 µM)	Targeted liposomes evidenced the immune system modulation and superior antitumor effect compared to all the other treatments.	[99]
		In vivo: female C57BL/6 mice; s.c. injection of B16OVA cells	i.v. injection of free DOX, non-targeted DOX liposomes, non-targeted DOX liposomes + free α-PD-L1 and Fab’-anti-PD-L1 DOX liposomes (3 mg/kg)		
Lecithin:SC:chol:peptide TD	Vemurafenib	In vitro: murine B16F10 and human A375 cell lines	Cell viability assay: MTT (0 to 50 μg/mL of vemurafenib and 0 to 1.25 mg/mL of lecithin)	The modification of vemurafenib liposomes with peptide TD potentiated the transdermal delivery of the compound, resulting in negligible adverse side effects compared to oral and i.v. routes.	[100]
		In vivo: male BALB/c nude mice; s.c. injection of A375 cells	oral, i.v. and transdermal administration of different formulations of vemurafenib (1.25 and 2.5 mg/mL)		
DOTAP:DOPE:chol:PEG	Anti-PD-1 siRNA	In vitro: murine B16F0 cell line	Cell viability assay: MTT (20 nM)	Anti-PD-1 siRNA liposomes demonstrated efficacy in silencing PD-1 mRNA expression in T cells, increasing the antitumor immune response.	[101]
		In vivo: female C57BL/6 mice; s.c. injection of B16F0 cells	i.v. administration of Doxil (5 and 10 mg/kg), liposomal scramble siRNA (5 µg/kg), liposomal siRNA (5 µg/kg), and liposomal siRNA (5 µg/kg) + Doxil (5 mg/kg)		
DOTAP:DOPE:C6-ceramide:SC	Curcumin and anti-STAT3 siRNA	In vitro: murine B16F10 cell line	Cell viability assay: MTT (250 μM of curcumin and 0.5 nM of siRNA)	Topical iontophoretic administration of a curcumin and anti-STAT3 siRNA nanosystem demonstrated similar tumor inhibition efficacy as observed for i.t. injection, but significantly higher compared to liposomes of each of the compounds individually by either route.	[102]
		In vivo: female C57BL/6 mice; s.c. injection of B16F10 cells	Topical (iontophoretic and passively) and i.t. administration of curcumin (3 mg/kg) and STAT3 siRNA (0.6 mg/kg) formulations alone or in combination		
DPPC:chol	5-ALA	In vitro: murine B16F10 cell line	Cell viability assay: WST-1 (630 nm; 50 J/cm^2^, 20 min) ROS detection assay: DHE Mitochondrial membrane potential assay: mitotraker	The combination of PDT and liposomes suggested greater phototoxicity compared to the non-liposomal form. In addition, in vivo assays demonstrated a higher ability to skin penetration in comparison to the free compound.	[103]
		In vivo: male BALB/c nude mice; s.c. injection of B16F10 cells	Topical administration of free 5-ALA and incorporated in a liposomal formulation for PDT application		
DPPC:MPPC:DSPE-PEG	ICG and poly I:C	In vivo: C57BL/6 mice; s.c. injection of B16F10 cells, followed by a metastatic model by i.v. injection of B16F10 cells	i.t. administration of empty liposomes, ICG loaded liposomes and co-loaded ICG and poly I:C liposomes for PTT application (808 nm; 1 W/cm^2^, 5 min)	When submitted to PTT, both ICG liposomes and liposomes co-loaded with ICG and poly I:C promoted anticancer effects. However, only the second formulation prevented lung metastases.	[91]

Abbreviations: 5-ALA, 5-aminolevulinic acid; CHEMS, cholesteryl hemisuccinate; Chol, cholesterol; DHE, dihydroethidium; DMPC, dimyristoyl phosphatidyl choline; DSPC, distearoyl phosphatidyl choline; DSPE-PEG, distearoyl phosphatidyl ethanolamine covalently linked to polyethylene glycol-2000; DOPE, dioleoyl phosphatidyl ethanolamine; DOTAP, dioleoyl trimethyl ammonium propane; DOX, doxorubicin; DPPC, dipalmitoyl phosphatidyl choline; Fab’, monovalent-variable fragments; HSPC, hydrogenated soya phosphatidyl choline; IC_50_, half-inhibitory concentration; ICG, indocyanine green; i.t., intratumoral; i.v., intravenous; MPPC, monopalmitoyl phosphatidyl choline; MTS, dimethylthiazol carboxymethoxyphenyl sulfophenyl tetrazolium; MTT, dimethylthiazol diphenyl tetrazolium bromide; PD-1, programmed cell death protein-1; PD-L1, programmed death ligand-1; PDT, photodynamic therapy; PEG, polyethylene glycol-2000; PI, propidium iodide; poly I:C, polyinosinic:polycytidylic acid; PTT, photothermal therapy; ROS, reactive oxygen species; s.c., subcutaneous; SC, sodium cholate hydrate; siRNA, small interference RNA; SPC, soybean phosphatidyl choline; SRB, sulphorodamine B; STAT3, signal transducer and activator of transcription 3; WST-1, water soluble tetrazolium salt.

#### 3.1.2. Solid Lipid Nanoparticles

SLNPs comprise a colloidal drug-delivery system consisting of a solid lipid matrix stabilized by a mixture of surfactants or polymers. This lipid matrix is solid at room and body temperatures [108]. Mono-, di- or triglycerides, fatty acids, fatty alcohols, waxes, and sterols are lipids, and tween 80, lecithin and sodium glycolate are surfactants, commonly used in this type of carrier [109,110]. A low capacity of hydrophilic drug encapsulation, the inability to release the drug uniformly, and its leakage during storage as a result of a decrease in its solubility during the crystallization process, might limit the drug’s use [88,111,112].

Among the various studies reporting the application of this type of lipidic system in the treatment of melanoma [113,114,115,116,117,118,119,120,121], one example is the work developed by Goto et al. [113]. In this work, aluminum chloride phthalocyanine (ClAlPc), a highly hydrophobic photosensitizing agent, was incorporated into the SLNP formulation for further PDT application. This type of therapy is already approved for the treatment of several cancers and the association of photosensitizers with nanotechnology systems has been investigated [122,123]. Therefore, after optimization of the formulation, its in vitro cytotoxicity was evaluated on B16F10 murine melanoma cells in the absence of and after laser irradiation. In the absence of laser irradiation, both ClAlPc SLNPs and ClAlPc in the free form did not show any type of toxicity. After irradiation, a high phototoxicity was observed especially in the case of the nanoformulated compound. This phototoxicity is directly related to the applied energy.

Another example, is a delivery system for temozolomide (TMZ) prepared by Clemente and his colleagues [116]. After optimizing the production method of SLNPs containing TMZ (TMZ SLNPs), cytotoxicity and clonogenicity studies of empty SLNPs, free TMZ and TMZ SLNPs were performed in human (JR8 and A2058) and murine melanoma cell lines (B16F10). These in vitro studies demonstrated a higher cytotoxicity of TMZ SLNPs compared to the free drug. Furthermore, experiments on human umbilical vein endothelial cells have also demonstrated its potential in inhibiting angiogenesis in a concentration-dependent way. Finally, in vivo assays conducted in a xenograft model of C57BL/6J mice, upon inoculation of B16F10 murine melanoma cells, demonstrated the superior ability of TMZ SLNPs on preventing tumor growth progression when compared to animals treated with TMZ in free form or receiving empty SLNPs. No toxic effects or morphological changes in the histological analysis were observed. In addition, an increase in animal survival was also achieved, with 100% of mice treated with nanoformulation surviving until the end of the experiment compared to about 50% of the control group. Furthermore, from the analysis of the tumor samples, an increase in the expression levels of IL-17A as well as a decrease in the expression of the cell proliferation marker ki-67 were found compared to animals treated with non-formulated TMZ. IL-17A is a T cell-derived pro-inflammatory cytokine whose role in carcinogenesis is not fully understood, and even different functions are being related. The potentiation of antitumor immunity is one of the described roles [124,125]. In turn, the high expression of ki-67 is associated to a poor prognosis, reflecting tumor cell proliferation and growth [126,127,128]. Regarding in vivo anti-angiogenic activity, TMZ SLNPs decreased the density of tumor microvessels (assessed by CD31 staining of samples) compared to the control group. However, and contrary to what was observed in vitro, this effect on inhibiting new vascularization was practically similar to that observed in animals treated with free TMZ. Table 2 briefly describes previous and additional examples of in vitro and in vivo experiments.

**Table 2 pharmaceutics-14-01817-t002:** In vitro and in vivo assessment of the therapeutic potential of different SLNPs in melanoma models.

Nanosystem Composition	Compound(s)	Model(s)	Summary of Experimental Assays and Conditions	Main Conclusions	Reference
Glyceryl behenate, sorbitan isostearate and polyoxyethylene-40 hydrogenated	ClAlPc	In vitro: murine B16F10 cell line	Cell viability assay: MTT without (200 and 400 μg/mL) and with (400 μg/mL) application of PDT (670 nm; 0.5, 1 and 2 J/cm^2^)	ClAlP SLNPs demonstrated remarkable phototoxic effects on melanoma cells compared to free ClAlPc.	[113]
Compritol 888, poloxamer 188 and tween 80	Curcumin and resveratrol	In vitro: murine B16F10 and human SK-MEL-28 cell lines	Cell viability assay: MTT (0.1 to 60 μg/mL of curcumin and 0.03 to 20 μg/mL of resveratrol) Cell proliferation assays: IncuCyte (same concentrations of MTT) and ECIS (60 and 20 μg/mL of curcumin and resveratrol, respectively)	The combination of the two compounds, either in solution or included in SLNPs, reduced the cancer cells viability compared to their use alone.	[114]
Cetyl palmitate, tricaprin and pluronic F68	PTX and IR-780	In vitro: murine B16 cell line	Cell viability assay: CCK-8 solution (0.1 to 10 μg/mL of PTX and 0.067 to 6.67 μg/mL of IR-780) after single (808 nm; 1 W/cm^2^, 5 min) or dual PTT treatment (repeated 24 h later) ROS detection assay: H_2_DCFDA Cell death assay: annexin V-FITC/PI	The combination of PTX/IR-780 SLNPs concentrated in DMNs with a dual PTT treatment inhibited dramatically the tumor growth, and a 100% cure rate was achieved.	[115]
		In vivo: female C57 mice; s.c. injection of B16 cells	Administration of DMNs loading PTX and/or IR-780 SLNPs for single (808 nm; 1 W/cm^2^, 5 min) and dual (repeated 24 h later) PTT treatment		
Sodium behenate and PVA	TMZ-C12	In vitro: human JR8 and A2058 and murine B16F10 melanoma cell lines	Cell viability assay: WST-1 (5 to 50 μM) Clonogenic assay: crystal violet/methanol (5 to 50 μM)	TMZ SLNPs demonstrated their greater cytotoxicity and anti-angiogenic activity in vitro compared to free TMZ. In vivo performance in terms of tumor growth inhibition and animal survival was also improved.	[116]
		In vivo: female C57BL6/J mice; s.c. injection of B16F10 cells	i.v. injection of free TMZ, empty SLNPs and TMZ SLNPs (0.5 μmol/g)		
SLT, GMS, TPGS and tween 20	DHA-dFdC	In vitro: murine B16F10 cell line	Cell viability assay: MTT (0.0001 to 10 μM)	DHA-dFdC SLNPs increased the chemical stability, plasma half-life and cytotoxicity of the compound in melanoma cells. Also its in vivo antitumor efficacy was improved compared to the free compound.	[117]
		In vivo: female C57BL/6 mice; s.c. injection of B16F10 cells	i.v. injection of DHA-dFdC solution, empty SLNPs and DHA-dFdC SLNPs (50 mg/kg)		

Abbreviations: ClAlPc, aluminum chloride phthalocyanine; DHA-dFdC, docosahexaenoyl difluorodeoxycytidine; DMNs, dissolving microneedles; ECIS, electrical cell-substrate impedance sensing; GMS, glycerol monostearate; i.v., intravenous; MTT, dimethylthiazol diphenyl tetrazolium bromide; PDT, photodynamic therapy; PTT, photothermal therapy; PTX, paclitaxel; PVA, polyvinyl alcohol; s.c., subcutaneous; SLNP, solid lipid nanoparticle; SLT, soybean lecithin; TMZ-C12, lipophilic prodrug of temozolomide; TPGS, tocopheryl polyethylene glycol 1000 succinate; WST-1, water soluble tetrazolium salt.

#### 3.1.3. Nanostructured Lipid Carriers

NLCs are considered the second generation of lipid NPs as they emerged as a way to overcome the drawbacks associated with SLNPs. NLCs comprise a mixture of solid lipids and liquid lipids, of which isopropyl myristate or oleic acid are examples, resulting in an unstructured lipid matrix. This fact promotes a greater capacity for drug incorporation and more uniform release, as well as greater stability with minimization of drug leakage during storage [88,111,112,129].

Of the various published studies [130,131,132,133,134,135], Malta and co-workers proposed the encapsulation of a new compound, 1-carbaldehyde-3,4-dimethoxyxanthone, in NLCs for further topical application [130]. The compound, previously synthesized by the same group [136], was denominated LEM2. It is a potent activator of the TAp73 with recognized antiproliferative effect on melanoma cells. However, it has a low water solubility and, consequently, low bioavailability. Thus, the compound was encapsulated, and different cytotoxic studies were performed in the human melanoma cell line, A375. Furthermore, in order to confirm whether the nanoformulation did not interfere with the molecular mechanism of the compound, the cell cycle progression and the expression levels of TAp73 protein and other proteins involved (p21, PUMA, BAX and Bcl-2) were evaluated. The results showed that the nanoformulation loading 2 μM of LEM2 promoted a significant increase in cell death, with cell cycle arrest in the G2/M phase, when compared to the empty nanosystem. An increase in the expression levels of TAp73, p21, PUMA, BAX, and MDM2, simultaneous with the reduction of the anti-apoptotic protein Bcl-2, were also achieved. Nevertheless, the validation of the developed NLC formulation was not assessed in vivo.

Liu et al., also proposed this type of system for drug delivery of docetaxel (DTX) [131]. Once again, the nanoformulation of the compound aimed to overcome its lower aqueous solubility. First, the production method of the nanoformulation was optimized and the resulting nanoformulation was characterized. In addition, the sustained release profile over the time of DTX from NLC was verified in vitro, with about 77% of DTX having been released in the first 24 h and the remaining up to 96 h. Then, in vitro cytotoxicity of the developed DTX NLC and Duopafei (DTX associated with high concentrations of Tween 80) was assessed in murine melanoma B16 cells. The results demonstrated a significantly higher cytotoxicity of the nanoformulation compared to Duopafei. Finally, in vivo antitumor efficacy of two different dosages of DTX NLC (10 and 20 mg/kg) in comparison with Duopafei (10 mg/kg) were tested in a melanoma model in Kunming mice. In this regard, although the lower dose has already achieved antitumor efficacy superior to Duopafei, DTX NLC 20 mg/kg was able to further reduce tumor volume even compared to DTX NLC 10 mg/kg. Furthermore, a greater loss of body weight was observed in the group of animals treated with Duopafei compared to any of the tested DTX NLC, suggesting fewer side effects of the nanoformulations. Table 3 depicts the cited examples and few examples more.

**Table 3 pharmaceutics-14-01817-t003:** In vitro and in vivo assessment of the therapeutic potential of different NLCs in melanoma models.

Nanosystem Composition	Compound(s)	Model(s)	Summary of Experimental Assays and Conditions	Main Conclusions	Reference
Precirol ATO 5, OA and tween 80	LEM2	In vitro: human A375 cell line	Cell viability assays: SRB (0.010 to 5 μM) and trypan blue (1 and 2 μM) DNA damage assay: cell cycle arrest (PI) (1 and 2 μM)	LEM2-loaded NLCs potentiate in vitro cell death of cancer cells in a dose-dependent manner, increasing the percentage of cell cycle arrest in the G2/M phase.	[130]
Stearic acid, GMS, SLT, OA and pluronic F68	DTX	In vitro: murine B16 cell line	Cell viability assay: MTT (0.01 to 10 μM)	In vitro and in vivo assays demonstrated the greatest antitumor efficacy of DTX NLCs. In addition, lower in vivo side effects were achieved compared to duopafei.	[131]
		In vivo: female Kunming mice; s.c. injection of B16 cells	i.v. injection of duopafei (10 mg/kg) and DTX NLCs (10 and 20 mg/kg)		
Stearylamine, IPM, SLT, TPGS and pluronic F68	Tripterine	In vitro: murine B16BL6 cell line	Cell viability assay: MTT (2 to 10 μg/mL) Cell uptake assay: HPLC	Cationic NLCs exhibited greater antitumor activity compared to neutral or anionic ones.	[132]
		In vivo: male C57BL/6 mice; s.c. injection of B16BL6 cells	Topical administration of compound solution, neutral, anionic and cationic NLCs (6 mg/kg) and i.p. injection of CTX as positive control (20 mg/kg)		

Abbreviations: CTX, cyclophosphamide; DLS, dynamic light scattering; DTIC, dacarbazine; DTX, docetaxel; GMS, glycerol monostearate; HPLC, high performance liquid chromatography; i.p., intraperitoneal; IPM, isopropyl myristate; i.v., intravenous; LEM2, carbaldehyde dimethoxyxanthone; MTT, dimethylthiazol diphenyl tetrazolium bromide; NLC, nanostructured lipid carrier; OA, oleic acid; PI, propidium iodide; s.c., subcutaneous; SLT, soybean lecithin; SRB, sulphorodamine B; TPGS, D-α-tocopheryl polyethylene glycol 1000 succinate.

### 3.2. Polymeric-Based Nanoparticles

The use of polymers has substantially increased in drug-delivery systems for the treatment of cancer [137]. Their easy production and surface modification, improved pharmacokinetic and pharmacodynamic characteristics, as well as their recognized stability, biocompatibility, and biodegradability are among the main reasons for their growing use [138,139,140,141]. In this context, the polymers commonly used can be divided into three main groups, depending on their origin: from natural sources, such as chitosan, starch, alginate, cellulose, hyaluronic acid, chondroitin sulfate, dextran or albumin; biosynthesized, such as poly β-hydroxybutyrate (PHB); or chemically synthesized, examples of which are polylactic (PLA), poly(lactic-glycolic acid) (PLGA), polyurethane (PU), polymethyl methacrylate resin (PMMA) or poly(ε-caprolactone) (PCL) [142,143,144]. Relevant data are depicted in Table 4.

Among the various polymers used for designing NPs for the treatment of melanoma [145,146,147,148,149,150,151,152,153,154,155,156], the natural polymer chitosan (CS) was proposed by Ferraz’s research team for the encapsulation of S-nitrosomercaptosuccinic acid (S-nitroso-MSA) [145]. The work carried out focused on studying in vitro the cell death mechanism associated with the formulation (S-nitroso-MSA-CS). For this, a B16F10 murine melanoma cell line was used. Regarding cytotoxic assays, a greater impact of S-nitroso-MSA-CS in reducing cell viability was found compared to empty NPs, non-nitroso MSA NPs or free S-nitroso-MSA, which did not demonstrate important cytotoxicity. Interestingly, preferential selectivity was observed toward the cancer cells tested (B16F10) rather than healthy cells (Melan A). Moreover, flow cytometry showed a large percentage of cells in late apoptosis. In parallel, increased caspase 3 activity, as well as increased cell viability, was observed upon incubation of cells with a caspase inhibitor (Boc-D-FMK). The integrity of the cell membrane was also verified by the absence of LDH release and inhibitors of necroptosis and necrosis were tested, not resulting in a decrease in cytotoxicity. Thus, taking all these results together, a caspase-dependent apoptotic mechanism is suggested. Likewise, S-nitroso-MSA-CS also demonstrated ability to increase reactive cellular and mitochondrial oxygen species compared to control. Additionally, it was also proven that the cytotoxic effect of the formulation is not due to the release of nitric oxide (^•^NO), but due to the direct transfer of the S-nitroso group present in the formulation to free thiol groups of proteins.

PLGA is a copolymer approved by the FDA due to its well-known properties [157,158]. Zhang et al. [152] developed a formulation of polymeric NPs composed of PLGA and poloxamer 407 for the controlled release of apatinib (Apa-PLGA NPs). Apatinib is a small-molecule inhibitor of angiogenesis, acting by supplying oxygen and nutrients to the tumor microenvironment through antagonism of the vascular endothelial growth factor receptor 2 (VEGF-2). The comparative cytotoxicity of Apa in free form or incorporated in PLGA NPs was performed in an in vitro model by using murine melanoma cells (B16) at different concentrations and incubation times. Overall, the antiproliferative properties of the nanoformulation was superior in comparison to Apa in the free form. Subsequently, the antitumor effect of the nanosystem was evaluated in a C57BL/6 mice melanoma model. For this, a first screening to evaluate the intratumoral dose of apatinib was performed. As a result, 6 mg/kg was the dose of apatinib chosen for incorporation into NPs and tested in another group of animals. The results showed that Apa-PLGA NPs promoted the highest reduction on tumor growth progression among all tested animal groups. Statistically significant differences were observed in terms of tumor regression between animals treated with Apa-PLGA NPs or receiving free apatinib. These results were also confirmed by histological analyses which demonstrated higher necrosis levels. In addition, a Western blot analysis detected a reduction in phosphorylation levels of VEGFR-2 and ERK1/2 as well as a reduction in VEGFR-2 protein levels in tumor tissues.

On the other hand, phototherapies such as PTT, in addition to their recognized ability to promote tumor cell death, have also been associated with the development of immunogenicity which, due to not being robust enough for an effective systemic antitumor response, is often combined with other therapeutic strategies [159]. An example in this regard was the work developed by other researchers who combined photothermal and epigenetic therapies in a single nanosystem [151]. The formulation comprises PLGA NPs containing, once again, the photothermal agent ICG and the epigenetic drug Nexturastat A (NextA). The antitumor and immunomodulatory activity of Next A, a histone deacetylase (HDAC) 6 inhibitor, supports its usage. First the photocytotoxic activity of the formulation on the murine melanoma cell line SM1 was evaluated in vitro. A reduction after PTT application of approximately 80% in cell viability were found for an ICG dose of 2 mg/mL. In addition, the effectiveness of inhibiting HDAC 6 by the formulation was also confirmed in SM1 and B16F10 murine cancer cell lines. In parallel, another important factor was the increase in the expression of the major histocompatibility complex Class I, typically downregulated in cancer cells, as well as costimulatory molecules. Lastly, a syngeneic murine melanoma model was established through inoculation of SM1 cells. The animals were distributed in six groups combining both therapies or not. The results showed that the combined therapy allowed to slow tumor progression and increase the survival of the animals. However, it was found that therapeutic efficacy is largely supported by the initial treatment of PTT, with epigenetic therapy not being sufficient to sustain these initially obtained anticancer effects over time. Table 4 summarizes the previous examples and the most representative ones.

**Table 4 pharmaceutics-14-01817-t004:** In vitro and in vivo assessment of the therapeutic potential of different polymeric-based nanoparticles in melanoma models.

Nanosystem Composition	Compound(s)	Model(s)	Summary of Experimental Assays and Conditions	Main Conclusions	Reference
Chitosan	S-nitroso-MSA	In vitro: murine B16F10 cell line	Cell viability assays (5, 10, 20, and 40 μg/mL): MTT, trypan blue and LDH release ROS detection assay: CM-H_2_DCFDA and MitoSOX Red Cell death assays: annexin V-FITC/PI and caspase-3 activity	Nanoformulation exhibited high cytotoxicity selectively on cancer cells.	[145]
PLGA and PVA	Xanthohumol	In vitro: murine B16F10 cell line	Cell viability assay: MTT (2 to 40 μM) Cell proliferation assay: wound healing	Loaded PLGA NPs showed high cytotoxicity as well as inhibition of proliferation and migration.	[146]
PMMA and sodium lauryl sulfate	α-terpineol	In vitro: murine B16F10 and human SK-MEL-28 cell lines	Cell viability assay: MTT (5, 50 and 500 μg/mL)	Nanosystem exhibited a large and selective cytotoxic effect in both melanoma cell lines tested.	[149]
PLA and PVA	DTIC and zinc phthalocyanine	In vitro: human MV3 cell line	Cell viability assay: MTT (20 and 100 μg of DTIC) after PDT application (660 nm; 28 J/cm^2^, 2.5 min)	In vitro assays demonstrated the added value of combined therapy in reducing cancer cell viability.	[150]
PLGA and PVA	ICG and NextA	In vitro: murine SM1 and B16F10 cell lines	Cell viability assay: Cell Titer-Glo ATP (0.5 to 2.0 mg/mL of NPs) with and without application of PTT HDAC activity assay: HDAC-Glo I/II	The combination of photothermal and epigenetic therapies increased the in vitro expression of immunological markers. Moreover, in an in vivo context, a delayed tumor progression and an improved median survival were achieved.	[151]
		In vivo: female C57BL/6 mice; s.c. injection of SM1 cells	i.t. administration of different formulation combinations (50 mg/kg of NPs) followed or not by PTT application (808 nm; 0.4 W, 10 min)		
PLGA and poloxamer 407	Apatinib	In vitro: murine B16 cell line	Cell viability assay: CCK-8 solution (4, 20 and 40 μg/mL)	In vitro and in vivo experiments demonstrated the high performance of Apa-PLGA NPs.	[152]
		In vivo: male C57BL/6 mice; injection of B16 cells	i.t. administration of free apatinib at different concentrations (2, 4 and 6 mg/kg), empty PLGA NPs and Apa-PLGA NPs (6 mg/kg)		
PCL, span 80, caprylic/caprictriglycerides and polysorbate 80	Resveratrol	In vitro: murine B16F10 cell line	Cell viability assay: MTT (1, 3, 10, 30, 100 and 300 μM)	Confirming the in vitro cytotoxicity results, the in vivo study demonstrated an increase in areas of inflammation and necrosis as well as a reduction of metastases and pulmonary hemorrhage compared to the free compound.	[153]
		In vivo: male and female C57BL/6J mice; s.c. injection of B16F10 cells	i.p. administration of free resveratrol, empty PCL NPs and resveratrol-PCL NPs (5 mg/kg)		
Chitosan, sodium alginate and calcium chloride	DOX	In vitro: murine B16F10 and B16OVA cell lines	Cell viability assay: alamar blue solution (1 to 100 μM)	In vitro assays suggested a greater intracellular accumulation and cytotoxicity of the nanosystem compared to the free drug. However, a similar effect between both was observed in the in vivo inhibition of tumor progression.	[154]
		In vivo: female C57BL/6 mice; s.c. injection of B16OVA cells	i.v. injection of free DOX, empty NPs and DOX NPs (3 mg/kg)		

Abbreviations: Apa, apatinib; CCK-8, cell counting kit-8; CM-H_2_DCFDA, chloromethyl dichlorodihydrofluorescein diacetate; DOX, doxorubicin; DTIC, dacarbazine; ICG, indocyanine green; FITC, fluorescein isothiocyanate; HDAC, pan-histone deacetylase; i.p., intraperitoneal; i.t., intratumoral; i.v., intravenous; LDH, lactate dehydrogenase; MTT, dimethylthiazol diphenyl tetrazolium bromide; NextA, nexturastat A; NPs, nanoparticles; PCL, poly(*ε*-caprolactone); PDT, photodynamic therapy; PI, propidium iodide; PLA, polylactic acid; PLGA, poly lactic-co-glycolic acid; PMMA, poly(methyl methacrylate); PTT, photothermal therapy; PVA, polyvinyl alcohol; s.c., subcutaneous; S-nitroso-MSA, S-nitrosomercaptosuccinic acid.

### 3.3. Metallic-Based Nanoparticles

In recent years, metallic NPs have been attracting the attention of researchers around the world due to their varied unique properties, whether physical, chemical, optical, magnetic, catalytic, or electrical. Furthermore, their biocompatibility, as well as their ease of synthesis and chemical modification are key factors [160,161,162]. Among the metals commonly used in the production of this type of NPs, some stand out: gold (Au), silver (Ag), iron (Fe), copper (Cu), cerium (Ce), platinum (Pt), titanium (Ti), and zinc (Zn) [163,164]. Furthermore, their oxides, hydroxides, sulfides, phosphates, fluorides, and chlorides can be considered [164]. Relevant in vitro and in vivo data are included in Table 5.

There are many studies exploring metallic nanosystems as a strategy for the treatment of melanoma [165,166,167,168,169,170,171,172,173,174,175,176]. An example are the cuprous oxide nanoparticles (Cu_2_O NPs) developed by Wang and colleagues [171]. Throughout the work, the authors sought to evaluate the antitumor efficacy of the formulation both in in vitro and in vivo melanoma models, as well as to understand the influence of some factors in cell death. First, in vitro cytotoxicity assays in a melanoma cell line (B16F10) showed the dose and time-dependent ability of Cu_2_O NPs to reduce cell viability. Based on a previous study performed by the same group [177], selectivity for cancer cells was observed. Furthermore, an apoptotic mechanism and significant inhibition of cancer cell migration and invasion was observed when compared to the control. Then, two in vivo melanoma models (subcutaneous and metastatic lung melanoma), through the inoculation of B16F10 cells, were performed. In the first, a significant inhibition of tumor progression of the group treated with Cu_2_O NPs compared to the control group was observed, leading to a significantly higher survival rate at the end of the experiment than the control group. In its turn, in the metastatic model, the formulation promoted a significant decrease of lung tumor nodules compared to the control group. Moreover, the safety and rapid, essentially hepatic clearance of the formulation was also confirmed in vivo. Finally, in mechanistic terms, it has been proven by the various assays that the cytotoxicity of Cu_2_O NPs, among other possible factors, is triggered by mitochondrial injury, which is reflected in a decrease of their membrane potential, increase of ROS, release of cytochrome C and activation of caspases 3 and 9.

Au NPs are another example of metallic NPs that have often been used, for example, in PTT [178,179,180,181] or for conjugation with drugs, as explored by other researchers [172,182,183]. Once again, and taking into account that different nanocarriers promote different pharmacokinetic and pharmacodynamic profiles, and consequently different accumulations sites [184,185], drug delivery of doxorubicin was tested, this time associated with Au NPs. The antitumor efficacy of ultra-small Au NPs (~3 nm) conjugated with doxorubicin (Au-DOX) compared to the use of DOX alone was evaluated. In one of their first studies, the ability of this conjugate to reduce IC_50_ value by a twenty-fold factor in DOX-resistant B16 cells was demonstrated, compared to DOX in its free form [182]. Later, in another study [183], the conclusion was also drawn that the impact that Au-DOX exerts on the malignant cell death is not due to the release of DOX as commonly observed in other approaches, but by the binding of the intact conjugate to cell structures. In its turn, in a more recent study published by the same group [172], the cell viability reduction previously observed for B16 cells, was not observed for the SK-MEL-28 cell line, a cell line sensitive to DOX. However, a two-fold factor reduction of cell viability was still achieved. In addition, here, two different in vivo melanoma models were established in C57BL/6 and nude mice, upon inoculation of a DOX-resistant murine cell line (B16) and a doxorubicin-sensitive human cell line (SK-MEL-28), respectively. For this, the animals were divided into different groups: control, Au NPs, DOX and Au-DOX. Analysis of the results from both in vivo models reveals the greater potential of Au-DOX in inhibiting the progression of tumor volume in a sustained manner over the time. Taking together the histology, as well as the TUNEL staining of the tumor samples from the two in vivo experiments, a greater necrotic component can be seen compared to apoptosis. The treatment effects were more strongly evidenced in the groups of animals submitted to the formulation under study.

In the case of PTT, Pandesh and his team synthesized Fe_3_O_4_ NPs surrounded by a gold shell (Fe-Au NPs) [174]. Although the Fe_3_O_4_ core gives the NPs the ability to magnetically target the tumor site through a magnet placed under the skin in the tumor region, the gold shells work as a photothermal agent for later application of PTT. First, the thermal conversion capacity of the formulation was tested by using different gold concentrations and laser power densities. The increase in temperature was higher in accord with a higher rise in gold concentration and laser power. Finally, the therapeutic efficacy of the formulation was evaluated through an in vivo model of melanoma. For this, C57BL/6 animals were subcutaneously inoculated with B16F10 cells for primary tumor development. When the desired volume was reached, five treatment groups were established: control, intravenous administration of Fe-Au NPs, intravenous administration of Fe-Au NPs plus laser and intravenous administration of Fe-Au NPs plus magnet plus laser. After two weeks, the mean tumor volume of animals that received the complete treatment (NPs + magnet + laser) increased 7.7-fold compared to 47.3-fold in animals that received no treatment (control group). Thus, an average tumor progression inhibition rate of 83.5% versus control was obtained for this group. The same treatment but in the absence of magnetic targeting of the NPs lowered the inhibition rate to 57%. These and more in vitro and in vivo experiments are briefly described in Table 5.

**Table 5 pharmaceutics-14-01817-t005:** In vitro and in vivo assessment of the therapeutic potential of different metallic-based nanoparticles in melanoma models.

Nanosystem Composition	Compound(s)	Model(s)	Summary of Experimental Assays and Conditions	Main Conclusions	Reference
MoO_3_ NPs	Non-applicable	In vitro: human G361 cell line	Cell viability assay: MTT (50–400 μg/μL)	MoO_3_ NPs showed selective cytotoxicity against malignant skin cells compared to healthy cells.	[165]
AgPt NPs	Non-applicable	In vitro: human A375 cell line	Cell viability assays: MTS (10–250 µg/mL)	NPs demonstrated antitumor activity in the A375 cell line while being safe for healthy cells.	[166]
Pd NPs	Non-applicable	In vitro: human A375 cell line	Cell viability assays (0–40 μg/mL): MTT and NRU DNA damage assays: comet, cell cycle arrest (EtBr) ROS detection assay: H_2_DCFDA oxidative stress detection assay: BCA Cell death assays: DAPI, AO/EtBr and caspase-3 activity	In vitro assays demonstrated cytotoxic and genotoxic activity of Pd NPs on melanoma cells.	[169]
Cu NPs	Non-applicable	In vitro: human A375 cell line	Cell viability assay: MTT (up to 4.5 μg/mL) Cell membrane fluidity assay: TMA-DPH DNA damage assays: comet (EtBr), chromosomal condensation (DAPI) and cell cycle arrest (PI) ROS detection assay: H_2_DCFDA Mitochondrial membrane potential assay: JC-1 Cell death assays: annexin V-FITC/PI and caspase-3 activity	Cu NPs promoted DNA damage, cell cycle arrest in the G2/M phase and depolarization of mitochondrial membrane potential.	[170]
Cu_2_O NPs	Non-applicable	In vitro: murine B16F10 cell line	Cell viability assay: MTT (2.5, 5 and 10 μg/mL) ROS detection assay: H_2_DCFDA Mitochondrial membrane potential assay: JC-1 Cell death assays: annexin V-FITC/PI and caspase-3 and caspase-9 activities	Cu_2_O NPs reduced selectively cancer cell lines viability in vitro. Similarly, in an in vivo context, a significant anti-tumor efficacy, impaired tumor growth progression and inhibition of lung metastasis was observed.	[171]
		In vivo: subcutaneous and metastatic models; male C57BL/6 mice; s.c. and i.v. injection of B16F10 cells, respectively	i.v. and i.t. administration of Cu_2_O NPs in subcutaneous (16 mg/kg) and metastatic (2 mg/kg) model, respectively		
Au NPs	DOX	In vitro: murine B16 and human SK-MEL-28 cell lines	Cell viability assay: SRB	Au-DOX was efficiently internalized and demonstrated great cytotoxicity in melanoma cells. In vivo assays also demonstrated its sustained inhibition of tumor progression over time when compared to DOX alone.	[172]
		In vivo: male C57BL/6 mice; s.c. injection of B16 cells and female nude mice, s.c. injection of SKMEL-28 cells	Administration of free DOX, Au NPs and Au-DOX conjugation (100 μM of DOX or 4 μL of Au)		
Ag and TiO_2_ based NPs	Non-applicable	In vitro: murine B16F10 cell line	Cell viability assay: MTT (75, 100, 150 and 200 μg/mL)	The formulation allied to PTT treatment markedly reduced tumor cells viability in vitro as well as the tumor volume in an in vivo model.	[173]
		In vivo: male and female C57BL/6J mice; s.c. injection of B16F10 cells	i.t. administration of formulation (100 μg/mL) followed or not by PTT application (808 nm; 2 W/cm^2^, 1 min)		
Fe_3_O_4_ and Au based NPs	Non-applicable	In vivo: male C57BL/6 mice; s.c. injection of B16F10 cells	i.v. administration of NPs (150 μg Au/mL) with and without PTT treatment (808 nm; 2.5 W/cm^2^, 6 min)	The magnetically target NPs associated with PTT impaired significantly the tumor growth in comparison to control group.	[174]

Abbreviations: Ag, silver; AgPt NPs, silver and platinum nanoparticles; AO, acridine orange; Au, gold; Au NPs, gold nanoparticles; BCA, bicinchoninic acid; Cu NPs, copper nanoparticles; Cu_2_O NPs, cuprous oxide nanoparticles; DAPI, diaminidino phenylindole; DOX, doxorubicin; EtBr, ethidium bromide; Fe_3_O_4_, iron(II, III)oxide; FITC, fluorescein isothiocyanate; H_2_DCFDA, dichlorodihydrofluorescein diacetate; i.t., intratumoral; i.v., intravenous; JC-1, tetrachloro tetraethylbenzimidazolylcarbocyanine iodide; MoO_3_ NPs, molybdenum trioxide nanoparticles; MTS, dimethylthiazol carboxymethoxyphenyl sulfophenyl tetrazolium; MTT, dimethylthiazol diphenyl tetrazolium bromide; NPs, nanoparticles; NRU, neutral red uptake; Pd NPs, palladium nanoparticles; PI, propidium iodide; POR, porphyrin derivate—Tetrakis-5,10,15,20-(2,4-dihydroxyphenyl)porphyrin; ROS, reactive oxygen species; s.c., subcutaneous; SRB, sulphorodamine B; TiO_2_, titanium dioxide; TMA-DPH, trimethylamonium diphenyl hexatriene.

### 3.4. Hybrid Nanoparticles

Hybrid nanosystems have different organic and/or inorganic materials in their composition, which results in unique synergistic functional properties [186,187,188,189,190]. There are several studies that have been carried out by using hybrid NPs in the treatment of melanoma [191,192,193,194,195,196,197,198,199,200,201]. A selection of in vitro and in vivo studies is shown in Table 6. In the liposomes subsection, a study was reported describing the encapsulation of a metallic compound with cytotoxic properties (Cuphen) [98]. Following the notable results in terms of safety and efficacy obtained in vitro and in vivo, researchers from the same group proposed the association of metallic iron oxide NPs with this liposomal nanosystem [191]. The aim was to take advantage of the magnetic targeting of these NPs in order to potentiate the accumulation of the nanoformulation at the tumor microenvironment. Thus, the objectives of this work were to evaluate the influence that the combination of magnetic NPs with Cuphen would have on its cytotoxic properties, as well as to confirm the magnetic properties of the final formulation. The in vitro studies performed in murine (B16F10) and human (MNT-1) melanoma cells, confirmed the maintenance of the cytotoxic activity of Cuphen in the presence of NPs. In addition, and in order to ensure the safety of its parenteral administration, the absence of hemolytic activity of the formulation was also confirmed.

Another type of hybrid NP developed in the context of melanoma management was presented by Lopes and collaborators [197]. The purpose was the use of Au NPs coated with hyaluronic and oleic acids and conjugated with epidermal growth factor (EGF-conjugated HAOA-coated Au NPs) for PTT application. After in vitro assays demonstrating the safety of these Au NPs without laser irradiation, a melanoma xenograft model in severely immunocompromised hairless mice was developed upon inoculation of A375 human melanoma cells. The aim was to evaluate the therapeutic efficacy of in situ administration of Au in combination with near-infrared (NIR) laser irradiation. For this, the animals were divided into four groups (control group, treatment group only submitted to 5 min of laser irradiation; and two other treatment groups submitted to the administration of Au NPs followed by 5 or 10 min of laser irradiation). As a result, in situ administration of Au NPs followed by NIR laser irradiation for 5 min showed the highest tumor volume reduction up to 80%. Moreover, a formation of numerous necrotic foci by histological analysis of tumor samples was observed. Finally, no toxic effects were observed in the different excised organs nor in the skin exposed to the irradiation. Table 6 displays an overview of hybrid nanoparticles.

**Table 6 pharmaceutics-14-01817-t006:** In vitro and in vivo assessment of the therapeutic potential of different hybrid nanoparticles in melanoma models.

Nanosystem Composition	Compound(s)	Model(s)	Summary of Experimental Assays and Conditions	Main Conclusions	Reference
IO NPs loaded liposomes (DMPC: CHEMS:DSPE-PEG)	Copper (II) complex—Cuphen	In vitro: human MNT-1 and murine B16F10 cell lines	Cell viability assay: MTT of free Cuphen (0.5 to 7 μM), free IO NPs (1 to 7.5 mg/mL) and their combination (Cuphen at 1 and 5 µM and IO NPs at 2 mg/mL)	IO NPs did not influence the cytotoxicity of Cuphen and when loaded in liposomes the magnetic properties were verified.	[191]
Gold coated loaded liposomes (HSPC)	Curcumin	In vitro: murine B16F10 cell line	Cell viability assays: PI and MTT of free curcumin, curcumin-liposomes and curcumin-lip/Au NPs (100 μg/mL) after PTT (780 nm; 650 mW, 5 min) Cell uptake assay: DAPI	Curcumin was efficiently internalized by cancer cells when incorporated into the formulation. Furthermore, its adjuvant effect in combination with PTT was shown.	[192]
BSA coated Ag NPs	Non-applicable	In vitro: murine B16F10 cell line	Cell viability assay: WST-1 without (10^−8^ to 10^−2^ M of Ag) and with (2.7 × 10^−3^ M of Ag) application of PTT (690 nm; 0.8, 0.9 and 1 W, 10 min) ROS detection assay: H_2_DCFDA	The formulation demonstrated its cytotoxicity by increasing the generation of ROS, while also exhibiting its added value as a photothermal agent.	[194]
Chitosan coated loaded liposomes (DMPC:chol)	ICG	In vitro: murine B16F10 cell line	Cell viability assay: MTT of free ICG and chitosan coated or uncoated IGC-liposomes (40 μM) without and with application of PDT (785 nm; 100 mW/cm^2^, 2.5 min) Cell uptake assay: fluorescence intensity	Chitosan coated ICG-liposomes increased the cellular uptake of ICG and consequently its photocytotoxicity, compared to uncoated ones.	[195]
HAOA coated Au NPs	Non-applicable	In vitro: murine B16F10 cell line	Cell viability assay: MTT without (5, 30 and 60 μM) and with (5 μM) application of PTT (811 nm; 0.04 W/cm^2^, 3 min)	The laser activation of HAOA-coated Au NPs demonstrated a reduction on cancer cell viability compared to observed for healthy cells (HaCat).	[196]
EGF-conjugated HAOA coated Au NPs	Non-applicable	In vitro: murine B16F10 and human A375 cell lines	Cell viability assay: MTT (25 to 100 μM) without application of PTT	The safety of the formulation without laser irradiation was confirmed in vitro. In turn, in vivo experiments showed that 5 min of laser irradiation promoted the greatest tumor volume reduction, about 80%.	[197]
		In vivo: male hairless SCID mice; s.c. injection of A375 cells	i.t. injection of EGF-conjugated coated Au NPs (20 mg/kg) followed by NIR laser irradiation (811 nm; 2.5 W/cm^2^, 5 or 10 min)		
Liposome (EPC:chol:DDAB:DSPE-PEG) containing HSA-loaded NPs	CHL	In vitro: murine B16F10 cell line	Cell uptake pathway assay: coumarin-6	CHL-hybrid NPs exhibited a more pronounced antitumor effect and increased overall survival compared to all the other formulations.	[198]
		In vivo: male C57BL/6 mice; s.c. injection of B16F10 cells	i.v. injection of CHL solution, CHL-liposomes, CHL-Alb NPs and CHL-Alb/liposome hybrid NPs (5 mg/kg)		

Abbreviations: Ag NPs, silver nanoparticles; Au NPs, gold nanoparticles; BSA, bovine serum albumin; CHEMS, cholesteryl hemisuccinate; CHL, chlorambucil; Chol, cholesterol; DAPI, diaminidino phenylindole; DDAB, dimethyl dioctadecyl ammonium bromide; EGF, epidermal growth factor; EPC, egg phosphatidylcholine; DMPC, dimyristoyl phosphatidyl choline; DSPE-PEG, distearoyl phosphatidyl ethanolamine covalently linked to polyethylene glycol-2000; HAOA, hyaluronic and oleic acids; has, human serum albumin; HSPC, hydrogenated soya phosphatidyl choline; ICG, indocyanine green; IO NPs, iron oxide nanoparticles; i.t., intratumoral; i.v., intravenous; MTT, dimethylthiazol diphenyl tetrazolium bromide; NPs: nanoparticles; NIR, near-infrared; PDT, photodynamic therapy; PI, propidium iodide; PTT, photothermal therapy; ROS, reactive oxygen species; s.c., subcutaneous; SCID, severe combined immune-deficient; WST-1, water soluble tetrazolium salt.

### 3.5. Examples of Patented Nanomedicine Products

In general, the research and development (R&D) process of a medicine can take up to 20 years. Thus, patenting is one of the main tools that allows investors to safeguard their discoveries, at least for a few years [202,203]. Therefore, and as a result of all the aforementioned pre-clinical research carried out in the field of nanotechnology, numerous product patents have been applied for a wide range of cancers, including melanoma. Some of them are shown in Table 7.

### 3.6. Landscape of Clinical Trials Using Different Types of Nanosystems

Clinical research regarding the application of nanomedicine in the treatment of cutaneous melanoma has been evolving over the last several years. In this context, completed and ongoing clinical trials that explore different types of nanosystems, such as micro- and nanoemulsions, liposomes, polymeric and hybrid NPs, are presented in Table 8.

However, and despite all efforts made, no nanotechnology-based product has yet been officially approved for clinical use. In addition, and according to the ClinicalTrials.gov database, accessed on 20 July 2022, there are some clinical trials that were left incomplete, whose recruitment was withdrawn, suspended or terminated for various reasons. According to the same database, recruitment status of closed clinical trials can be classified into four categories: withdrawn, interrupted earlier, even before enrolling any participant; terminated, interrupted earlier and without possibility of restarted; suspended, interrupted earlier, but can be restarted; and completed, completed normally. In this context and accordingly, the information of public domain, phase 1/2 clinical trials, evaluating the pharmacodynamics and pharmacokinetics of liposomal miR-34 (MRX34) (NCT02862145), and a safety and efficacy study of the combination of nab-paclitaxel, temozolomide, and bevacizumab for the treatment of melanoma brain metastases (NCT02065466), are examples of withdrawn clinical trials. In this case, the published reasons were due to serious adverse immune-related side effects and lack of accrual, respectively. In addition to these, and by strategic decision, a phase 2 study which aimed at comparative analysis of the safety and efficacy of the combination of CC-486 (oral azacytidine) with nab-paclitaxel versus nab-paclitaxel alone (NCT01933061) is another example of a withdrawn clinical trial whose study object was nanotechnology in melanoma treatment. In turn, the phase 1 safety and efficacy study of liposomal cytarabine in combination with lomustine and brain radiation therapy for the treatment of leptomeningeal melanoma metastases (NCT01563614) was terminated due to the availability of other therapeutic approaches.

### 3.7. Regulation of Nano-Based Products

Despite the already wide range of nanodrugs approved for clinical use, regulatory policies remain non-harmonized across different geographic locations around the world [204,205]. It is true that nanomedicines have the same requirements in terms of quality, safety, and efficacy as other medicines. However, due to their small size and large surface area, they require specific considerations, especially with regard to their production and quality control, safety, and efficacy [206,207].

The lack of a unanimous definition or classification of the concept itself is a stumbling block. There is still to be highlighted a high demand in terms of establishing analytical methods depending on the nanomaterial used, as a result of its great heterogeneity, a divergence of the pharmacokinetic profiles from standardized constituent materials as well as the limitations of toxicity studies in assessing the short- and long-term impact in an in vivo situation. In addition, problems with reproducibility between batches, stability when scaling up or even the possible impact they might or might not have on the environment are also pressing issues [204,208,209].

However, there are a few guidelines, not legally binding, representing only their current opinion on the subject [205]. However, the increased number of nanopharmaceutical applications pending approval has pressured regulatory agencies to establish working groups to clearly determine and align some definitions and standards.

In the European Union, the legislation applied by the EMA to nanomedicines is the same applied to other medicines, and its approval or disapproval is supported by a risk/benefit analysis. There is a legal reference dated to 2011 that establishes the definition of nanomaterial (Recommendation 2011/696/EU) [210]. Furthermore, it should be noted that legally (REACH EC 1907/2006) [211], the safety of chemicals, including nanomaterials, is from the responsibility of the European Chemical Agency (ECHA). In the European context, the Nanomedicines Expert Group was one of the established groups, aiming the development of these types of subjects. This group founded projects such as the European Nanomedicine Characterization Laboratory (EUNCL), which is dedicated to the development of methods for the pre-clinical characterization of nanomedicines and the Regulatory Science Framework for Nano(bio)material-based Medical Products and Devices (REFINE), which focus its activity on the development of methods to support regulatory decisions [204,208,209].

Likewise, the FDA has not yet issued specific regulations with regard to nanomedicine. In 2017, a guide for the industry that includes biological products containing nanomaterials was made public; however, it is not legally binding [212]. Thus, approval requests made to the FDA are analyzed on a case-by-case basis and according to concrete product specifications through statutory and regulatory authorities. In addition, the FDA has mechanisms to assist industries in various practical issues during the development of nanodrugs as well as monitoring, soon after they enter the market. Among the groups created by the FDA to deal with underdevelopment in this regulatory area, there is the Nanotechnology Task Force and Nanotechnology Interest Group. Also other external institutes such as the Nanotechnology Characterization Laboratory of the National Cancer Institute (NCL-NCI) and the National Nanotechnology Initiative (NNI) have contributed to the development in the field of nanomedicine regulation [204,208,209].

Furthermore, other joint international efforts have also been made in this area. An example is the Global Summit on Regulatory Science: Nanotechnology Standards and Applications organized in 2019 by the Global Coalition for Regulatory Scientific Research (GCRSR). GCRSR comprises regulatory bodies from various countries who came together to discuss the gaps in nanomedicine regulation and ways to bridge them through global collaborations [207].

## 4. Conclusions and Future Perspectives

As previously described, cutaneous melanoma is a highly complex, unpredictable and heterogeneous malignancy, associated with increasing incidence and mortality rates. As reported throughout this review, since 2011, with the approval of immunotherapy and targeted therapy, there has been an authentic revolution in the treatment of melanoma with countless other drugs being approved since then. These advances have been crucial to improve the quality of life of patients, as well as to increase their average life expectancy. Even so, the results achieved are still inferior to those observed for other types of cancer, with heterogeneous responses and not always durable. In this context, nanomedicine emerges as a window of opportunities, in an attempt to improve response rates while reducing adverse side effects. In the last decades, nanotechnology has experienced great evolution and nowadays provides several tools capable of overcoming some of the obstacles associated with conventional therapies. Solving solubility and stability problems of several drugs in clinical use, increasing blood circulating half-lives and accumulation at tumor sites, by passive or active delivery, are some of the potentialities offered by this type of strategy. Its value is indisputable, and the proof of this are the nanomedicines already approved for a huge variety of pathologies including cancer, as well as the large number of clinical trials currently ongoing. In the specific case of melanoma, and although several nanosystems have been tested in different animal models, those that have reached clinical trials are so far predominantly liposomes and polymeric NPs. However, there are still pending issues to solve or clarify regarding the approval of novel nano-drug-delivery systems: How to assess potential risks of components of a nanosystem for human health in the short and/or long term? How to ensure reproducibility between batches? How to guarantee the stability of nanoformulations scale up? These are some of the questions that urgently need to be answered and on which the scientific community and regulatory agencies have to concentrate their efforts.

## Figures and Tables

**Figure 1 pharmaceutics-14-01817-f001:**
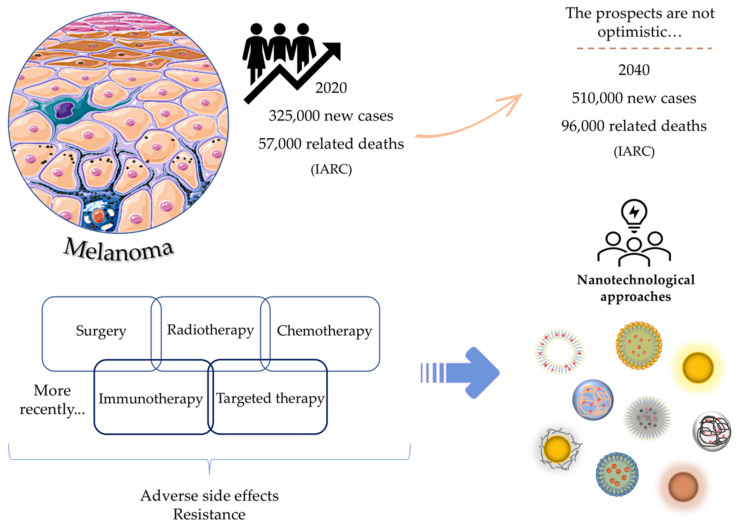
Melanoma: epidemiological data, current therapies and new alternative strategies for improving clinical outcomes.

**Figure 2 pharmaceutics-14-01817-f002:**
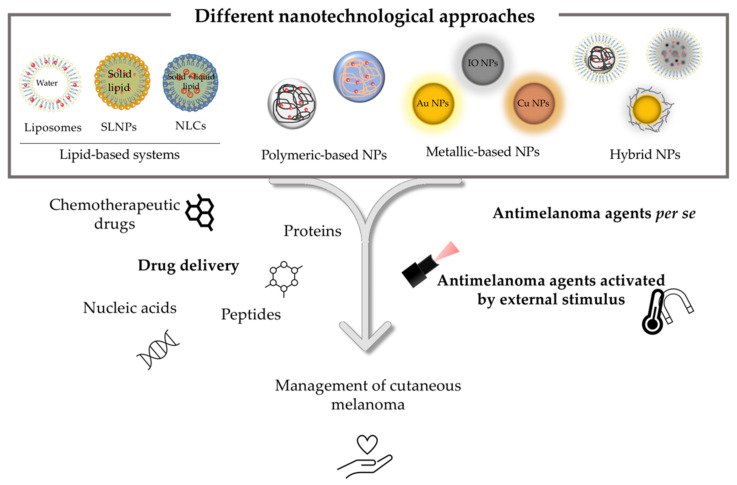
Schematic illustration of different types of nanosystems: lipidic, polymeric, metallic, and hybrids that can be used for loading chemotherapeutic drugs, proteins, peptides, nucleic acids, DNA, or for activation by external *stimuli* for melanoma management.

**Table 7 pharmaceutics-14-01817-t007:** Selected patented nanomedicine products with application to melanoma (not exclusively).

Nanosystem	Inventor(s) Name(s)	Grant Application Date	Patent Number
Drug loaded Fe(III)-DOPA NPs	Hyung Joon Cha, Bum Jin Kim and Ho gyun Cheong	June 2017	US9675629B2
Drug loaded PEG-PCL NPs	Adam W. G. Alani	July 2018	US10016422B2
Silicon dioxide NPs functionalized with an antigen	Markus Weigandt, Andrea Hanefeld, Armin Kuebelbeck and Gregor Larbig	October 2018	US10111952B2
Iron garnet NPs containing activatable nuclides	Anthony J. Di Pasqua, Kenneth J. Balkus, Imalka S. Munaweera and Yi Shi	February 2019	US10195297B2
PVP coated silver prussian blue NPs	Sudip Mukherjee and Chitta Ranjan Patra	March 2019	US10231996B2
Oil or water-in-oil emulsion combining antigens loaded liposomes	Pirouz M. Daftarian, Marc Mansour, Bill Pohajdak, Robert G. Brown and Wijbe M. Kast	April 2019	US10272042B2
T cell ligands and/or antigens linked to carbon nanotubes	Tarek M. Fahmy, Lisa D. Pfefferle, Gary L. Haller and Tarek R. Fadel	November 2019	US10485856B2
Complexes of albumin NPs and antibodies	Svetomir N. Markovic and Wendy K. Nevala	September 2020	US10765741B2
Drug loaded lipid-based NPs decorated with CD47	Raghu Kalluri and Sónia Melo	March 2021	US10959952B2
Sensitizer loaded PGA-based polymer/co-polymer/derivate NPs (PTT and SDT)	Nikolitsa Nomikou	June 2021	US11040101B2
Irinotecan-mesoporous silica NPs coated with a lipid bilayer	Andre E. Nel, Huan Meng and Xiangsheng Liu	August 2021	US11096900B2
Photosensitizer or drug loaded lipid layer coated NPs (PDT, but not only)	Wenbin Lin, Xiaopin Duan, Christina Chan and Wenbo Han	February 2022	US11246877B2
NIR absorbing dye based composite NPs (PTT)	Sehoon Kim, Youngsun Kim, Keunsoo Jeong and Gayoung Kim	April 2022	US11291726B2

Abbreviations: DOPA, 3,4-dihydroxyphenylalanine; NIR, near infrared; NPs, nanoparticles; PCL, poly(ε-caprolactone); PDT, photodynamic therapy; PEG, polyethylene glycol; PGA, polyglutamic acid; PTT, photothermal therapy; PVP, poly(n-vinyl-2-pyrrolidone); SDT, sonodynamic therapy. Data collected from the United States Patent and Trademark Office and Espacenet patent search websites, accessed on 20 July 2022.

**Table 8 pharmaceutics-14-01817-t008:** Examples of the most representative (not all) clinical trials (completed or ongoing) using different types of nanosystems for melanoma treatment.

Clinical Trial Phase	Clinical Trial Description	Melanoma Stage	Sponsor	Starting Date	Study Completion/Estimated Date	Trial ID
**Completed clinical trials**						
1	Pharmacokinetic study of a liposomal vincristine sulfate formulation.	III/IV	Acrotech Biopharma LLC	February 2005	November 2007	NCT00145041
	Safety and efficacy of a liposomal vaccine targeting dendritic cells (Lipovaxin-MM).	IV	Lipotek Pty Ltd.	September 2009	March 2012	NCT01052142
	Safety, pharmacokinetic and pharmacodynamic study of BIND-014 (PSMA-targeted PLA/PEG docetaxel NPs).	Advanced or metastatic	BIND Therapeutics	January 2011	February 2016	NCT01300533
2	Safety and efficacy of ABI-007 (nab-paclitaxel).	Unresectable locally recurrent or metastatic	Jonsson Comprehensive Cancer Center	February 2004	January 2010	NCT00081042
	Safety and efficacy of co-administration of ABI-007 (nab-paclitaxel) and carboplatin.	IV	Alliance for Clinical Trials in Oncology	October 2006	March 2010	NCT00404235
	Safety and efficacy analysis of the combination of nab-paclitaxel and bevacizumab versus ipilimumab alone.	IV	Academic and Community Cancer Research United	October 2013	October 2019	NCT02158520
3	Safety and efficacy of ABI-007 (nab-paclitaxel) versus dacarbazine.	IV	Celgene	April 2009	February 2014	NCT00864253
**Ongoing clinical trials**						
1	Safety and efficacy of nab-paclitaxel and bevacizumab.	IV	Mayo Clinic	March 2014	June 2025	NCT02020707
	Safety and tolerability of a cancer vaccine composed by naked RNA-drug products in a liposomal formulation (Lipo-MERIT).	IIIB/C/IV	BioNTech SE	March 2015	May 2023	NCT02410733
	Safety and tolerability of escalating doses of OX40L, IL-23 and IL-36γ encoding human mRNAs encapsulated in a lipid nanoparticle alone or combining with durvalumab.	Advanced or metastatic	ModernaTX, Inc.	November 2018	January 2023	NCT03739931
2	Safety and efficacy of the combination of nab-paclitaxel, carboplatin and endostatin after failure of PD-1 therapy.	Advanced	Peking University Cancer Hospital & Institute	March 2019	September 2022	NCT03917069

Abbreviations: Nab-paclitaxel, nanoparticle albumin-bound paclitaxel; PD-1, programmed cell death protein-1; PEG, polyethylene glycol; PLA, polylactic acid; PSMA, prostate-specific membrane antigen; TLR4, toll-like receptor 4. Data collected from the ClinicalTrials.gov database, accessed on 20 July 2022.

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
