# Peer review of "How to Treat Melanoma? The Current Status of Innovative Nanotechnological Strategies and the Role of Minimally Invasive Approaches like PTT and PDT"

_pharmaceutics, 2022, doi:10.3390/pharmaceutics14091817_

Round 1

Reviewer 1 Report

Generally, the authors review recent advances in nanotechnology for the treatment of melanoma. Different types of nanosystems are described, including lipid-based systems, polymeric-based nanoparticles, metallic-based nanoparticles, and hybrid nanoparticles. Besides, the status of nanodrugs for clinical use is introduced and further prospects are also discussed. Overall, this topic is of good interest to the readership of the journal, few issues should be settled before being accepted.

(1)    This manuscript focuses on nanotechnology-based therapy to treat melanoma. It is recommended to emphasize nanotechnology in the title

(2)    Line 130, “the nanosystems can even be activated by external stimulus, such as PTT and PDT” should be “the nanosystems can even be activated by external stimuli, such as light”

(3)    Some format issues in the reference section should be concerned. For example,

Ref 2, 18, 42, 69, 71, 176 , page information is missing

Ref 153, the author's name may be wrong

Author Response

Comments to the authors

Generally, the authors review recent advances in nanotechnology for the treatment of melanoma. Different types of nanosystems are described, including lipid-based systems, polymeric-based nanoparticles, metallic-based nanoparticles, and hybrid nanoparticles. Besides, the status of nanodrugs for clinical use is introduced and further prospects are also discussed. Overall, this topic is of good interest to the readership of the journal, few issues should be settled before being accepted.

Author response: The authors would like to express their profound gratitude to the reviewer for the constructive suggestions made to our article. We have carefully considered all comments, to which we have elaborated a point-by-point response. In the paper, all revised sections are   yellow highlighted

Questions to the authors

  1. This manuscript focuses on nanotechnology-based therapy to treat melanoma. It is recommended to emphasize nanotechnology in the title

Author response: Thanks for your suggestion. We appreciate and agree with referee’s point of view. The title has been updated accordingly.

“How to treat melanoma? The current status of innovative therapies and the role of minimally invasive approaches like PTT and PDT” was changed to:

“How to treat melanoma? The current status of innovative nanotechnological strategies and the role of minimally invasive approaches like PTT and PDT”

  1. Line 130, “the nanosystems can even be activated by external stimulus, such as PTT and PDT” should be “the nanosystems can even be activated by external stimuli, such as light”

Author response: Thank you for your comment. The alteration was made. In addition to this, in table 8,  few adjustments were made.

  1. Some format issues in the reference section should be concerned. For example,

Ref 2, 18, 42, 69, 71, 176 , page information is missing

Ref 153, the author's name may be wrong

Author response: Thank you for your remark. These and other missing informations were added, and the author's name in ref. 153 was corrected.

Once more, the authors thank Reviewer 1 for her/his valuable contribution to our article.

Reviewer 2 Report

1. English editing is strongly required.

2. The First mention is written in full and then can use abbreviations for the rest: NP (L24, 417, 507, 508), ROS (151), NLCs (322), NO (403), AuNPs (487, 489), I&D (570), EMA (631), etc.

Certain words are always in italics and should be corrected as such (eg In vitro, in vivo L460, L514, etc).

The physicochemical and not physica-chemical was previously used and should be corrected (L454).

3. What is/are the skin type most affected by melanoma?

4. The predominant use of murine cells is evident, what is the specificity of this model that justifies the preference or choice?

Both (3 and 4) can be included in section 2 and will definitely add value to the manuscript.

5. What are the principal factors for such prediction (L48)?

6. What is the revolution all about? (L60-61).

7. Shorten long sentences for clarity purposes (not concise nor clear, eg. L68-72), and many statements are to be rephrased (eg. L291-294, 301-304, 426-428, 511-513, etc). Certain words/expressions are repeated within the same sentences (eg. control group L474-475) and many more similar issues.

8. More descriptive legends are required for figs 1 and 2.

9. What are the functions of proteins IL17A and Ki67? please write short statements about their roles for clarifying and specifying their mention and choice.

10.  What are the clearance mechanisms for NPs systems (with limited biodegradability index)? What is the future of NPs that are not biodegradable nor cleared out of biological systems? How the rapid clearance was executed (L477)?

11. In Table 3, Characterization (encapsulation efficiency, drug release, and stability) cannot be considered as a study model but rather assessment parameters. Please correct with suitable information.

12. Cell death is a synergetic event resulting from more than one induced mechanism, but one or two might be predominant in many scenarios. Some parameters were tested in various studies to validate one or a couple of the mechanisms but cannot be considered solely as such to justify the resulting response/cell death. Therefore statements need to be accordingly revised (eg. L423, 466, 478-480).

13. North American (L430) or Canadians (L484), do geographical locations of the authors have a pivotal role in the study or narrative? If not, please revise.

14. When referring to Photothermal therapy (PTT), please specify the temperature difference, if possible.

15. What are the laser parameters (wavelengths, fluences/power/continuous or not, etc)? And those should be included where suitable?

16. Refs 6 and 7 are similar

Author Response

Author response: The authors would like to express their profound gratitude to the reviewer for the constructive suggestions made to our article. We have carefully considered all comments, to which we have elaborated a point-by-point response. In the paper, all revised sections are  yellow highlighted.

Questions to the authors

  1. English editing is strongly required.

Author response: Manuscript was revised in order to meet quality standards of the journal.

  1. The First mention is written in full and then can use abbreviations for the rest: NP (L24, 417, 507, 508), ROS (151), NLCs (322), NO (403), AuNPs (487, 489), I&D (570), EMA (631), etc.

Author response: Along all manuscript the abbreviations when appearing for the first time were fully written and, therefore, some adjustments were made.

  1. Certain words are always in italics and should be corrected as such (eg In vitro, in vivo L460, L514, etc).

Author response: We have checked all manuscript and we have changed accordingly. However, the italics for “in vitro” and “in vivo” are not used in this journal.

  1. The physicochemical and not physica-chemical was previously used and should be corrected (L454).

Author response: This sentence was corrected in the revised manuscript according to the referee’s suggestion.

  1. What is/are the skin type most affected by melanoma?

Author response: The skin types most affected by melanoma were added in section 2.

  1. The predominant use of murine cells is evident, what is the specificity of this model that justifies the preference or choice? Both (5 and 6) can be included in section 2 and will definitely add value to the manuscript.

Author response: The reply to comment 6 was included in section 3.1. “Most of in vitro studies use the murine cell line B16F10. This cell line exhibits a morphology of spindle-shaped and epithelial-like cells that was isolated from skin tissue of a mouse with melanoma. These cells are highly metastatic and present strong pigmentation (1). Furthermore, in vivo studies are most of the time tested in C57BL/6 male mice, that is the mice strain in which the melanoma cell line B16F10 was created (2). This strain is a widely used in immunocompetent murine melanoma models (3).

1.Predoi, M.-C.; Mîndrila, I.; Buteica, S.A.; Marginean, O.M.; Mîndrila, B.; Niculescu, M. Pigmented melanoma cell migration study on murine syngeneic B16F10 melanoma cells or tissue transplantation models. J. Mind Med. Sci. 2019, 6, 327–333. [CrossRef]

2.Fidler, I.J.; Nicolson, G.L. Organ selectivity for implantation survival and growth of B16 melanoma variant tumor lines. J. Natl. Cancer Inst. 1976, 57, 1199–1202. [CrossRef] [PubMed]

3. Coricovac, D.; Dehelean, C.; Moaca, E.A.; Pinzaru, I.; Bratu, T.; Navolan, D.; Boruga, O. Cutaneous melanoma—A long road from experimental models to clinical outcome: A review. Int. J. Mol. Sci. 2018, 19, 1566. 

  1. What are the principal factors for such prediction (L48)?

Author response: The main factors for the increase in new cases and deaths of melanoma were added to the introduction section in the revised version of the manuscript - Lines 56-58 of the revised version.

  1. What is the revolution all about? (L60-61).

Author response: The term ”revolution” was literally applied to emphasize new and more effective molecules used for melanoma treatment.

  1. Shorten long sentences for clarity purposes (not concise nor clear, eg. L68-72), and many statements are to be rephrased (eg. L291-294, 301-304, 426-428, 511-513, etc). Certain words/expressions are repeated within the same sentences (eg. control group L474-475) and many more similar issues.

Author response: The English was revised in all manuscript, accordingly to the high-quality standard of the journal.

  1. More descriptive legends are required for figs 1 and 2.

Author response: The legends of figures 1 and 2 were changed accordingly.

  1. What are the functions of proteins IL17A and Ki67? please write short statements about their roles for clarifying and specifying their mention and choice.

Author response: The role of proteins IL-17A and Ki-67 were included in the revised version of the manuscript.

  1. What are the clearance mechanisms for NPs systems (with limited biodegradability index)? What is the future of NPs that are not biodegradable nor cleared out of biological systems? How the rapid clearance was executed (L477)?

Author response: The topic suggested was added to the manuscript.

  1. In Table 3, Characterization (encapsulation efficiency, drug release, and stability) cannot be considered as a study model but rather assessment parameters. Please correct with suitable information.

Author response: The first example in Table 3 was removed.

  1. Cell death is a synergetic event resulting from more than one induced mechanism, but one or two might be predominant in many scenarios. Some parameters were tested in various studies to validate one or a couple of the mechanisms but cannot be considered solely as such to justify the resulting response/cell death. Therefore statements need to be accordingly revised (eg. L423, 466, 478-480).

Author response: The statements reported were revised according to referee’s suggestion.

  1. North American (L430) or Canadians (L484), do geographical locations of the authors have a pivotal role in the study or narrative? If not, please revise.

Author response: The reference to the geographical locations of the author was eliminated.

  1. When referring to Photothermal therapy (PTT), please specify the temperature difference, if possible.

Author response: The temperature range associated with photothermal therapy was added in the revised version of the manuscript.

  1. What are the laser parameters (wavelengths, fluences/power/continuous or not, etc)? And those should be included where suitable?

Author response: Thanks for your suggestion. Information provided in each phototherapy article about laser parameters was added throughout the tables.

  1. Refs 6 and 7 are similar.

Author response: Thanks for your observation, but these 2 papers refer to different subjects. Reference 6 refers to the number of new melanoma cases predicted for 2020 and 2040, while data in reference 7 is about the predicted number of melanoma deaths for the same years.

Once more, the authors thank Reviewer 2 for her/his valuable contribution to our article.

Reviewer 3 Report

The author starts with the refractory and high mortality of melanoma, firstly introduces the current mainstream traditional treatment methods, and then introduces the progress of the treatment of melanoma in the field of nanotechnology in the past 10 years. Challenges in translating nanotechnology-based therapies into clinical applications are also discussed. However, minor revision should be done to further improve the quality of this manuscript.

1. There are inaccurate expressions in some places in the text, such as section 3.1. "Lipid-based drug delivery systems have been gaining more and more notoriety by improving currently available cancer therapies", please check carefully and make corrections.

2. The summary of various drug-carrying carriers in this paper is a simple data description, and there is little description of the interrelated mechanisms. Can you choose the key points and describe them in detail?

3. Typical reviews on photothermal therapy should be cited and discussed (e.g., Nanoscale, 2019, 11(34): 15685-15708; Advanced Materials, 2020, 32(13): 1902333; Coordination Chemistry Reviews, 2021, 430: 213662; Coordination Chemistry Reviews, 2020, 419: 213393).

4. The Conclusion and future perspectives section in the article is too simple, and does not provide a complete and detailed summary of nanotherapy for melanoma. Please add it appropriately.

5. Some relevant references should be cited (e.g., Applied Materials Today, 2020, 18: 100464; Chemistry of Materials, 2019, 31(16): 6174-6186; Applied Materials Today, 2020, 18: 100464; iScience, 2020, 23(7): 101281.).

Author Response

The author starts with the refractory and high mortality of melanoma, firstly introduces the current mainstream traditional treatment methods, and then introduces the progress of the treatment of melanoma in the field of nanotechnology in the past 10 years. Challenges in translating nanotechnology-based therapies into clinical applications are also discussed. However, minor revision should be done to further improve the quality of this manuscript.

Author response: The authors would like to express their profound gratitude to the reviewer for all constructive suggestions made to our article. Each of the comments deserved our best attention, and a point-by-point response was elaborated. In the resubmitted version of the manuscript, all revised sections are yellow highlighted.

Questions to the authors

  1. There are inaccurate expressions in some places in the text, such as section 3.1. "Lipid-based drug delivery systems have been gaining more and more notoriety by improving currently available cancer therapies", please check carefully and make corrections.

Author response: Thank you for your comment. The sentence "Lipid-based drug delivery systems have been gaining more and more notoriety by improving currently available cancer therapies", was changed to:

“Lipid-based drug delivery systems are gaining more and more notoriety, either by improving currently available therapies, or by creating new ones”.

  1. The summary of various drug-carrying carriers in this paper is a simple data description, and there is little description of the interrelated mechanisms. Can you choose the key points and describe them in detail?

Author response: New data was added, strictly following the Referee’s suggestions.

  1. Typical reviews on photothermal therapy should be cited and discussed (e.g., Nanoscale, 2019, 11(34): 15685-15708; Advanced Materials, 2020, 32(13): 1902333; Coordination Chemistry Reviews, 2021, 430: 213662; Coordination Chemistry Reviews, 2020, 419: 213393).

Author response: Thank you for your feedback and literature suggestions. We have included the references aforementioned in the revised version.

  1. The Conclusion and future perspectives section in the article is too simple, and does not provide a complete and detailed summary of nanotherapy for melanoma. Please add it appropriately.

Author response: Thank you for your feedback. The conclusions and future perspectives section were  updated and expanded in order to include some of the benefits and added value that nanotechnology has brought to medicine, specifically, in improving the outcome of melanoma treatments.

  1. Some relevant references should be cited (e.g., Applied Materials Today, 2020, 18: 100464; Chemistry of Materials, 2019, 31(16): 6174-6186; Applied Materials Today, 2020, 18: 100464; iScience, 2020, 23(7): 101281.).

Author response: Thank you for your comment and literature suggestions. The recommended references were now included in the revised manuscript.

Once more, the authors thank Reviewer 3 for her/his valuable contribution to our article.

Round 2

Reviewer 3 Report

The authors have made very detailed revisions, and I recommend this paper for publication in Pharmaceuticals.